# Demographics, Clinical Characteristics and Survival Outcomes of Primary Urinary Tract Malignant Melanoma Patients: A Population-Based Analysis

**DOI:** 10.3390/cancers15184498

**Published:** 2023-09-10

**Authors:** Simone Morra, Reha-Baris Incesu, Lukas Scheipner, Andrea Baudo, Letizia Maria Ippolita Jannello, Mario de Angelis, Carolin Siech, Jordan A. Goyal, Zhe Tian, Fred Saad, Gianluigi Califano, Roberto la Rocca, Marco Capece, Shahrokh F. Shariat, Sascha Ahyai, Luca Carmignani, Ottavio de Cobelli, Gennaro Musi, Derya Tilki, Alberto Briganti, Felix K. H. Chun, Nicola Longo, Pierre I. Karakiewicz

**Affiliations:** 1Cancer Prognostics and Health Outcomes Unit, Division of Urology, University of Montréal Health Center, Montréal, QC H2X 3E4, Canada; re.incesu@uke.de (R.-B.I.); l.scheipner@medunigraz.at (L.S.); andrea.baudo@unimi.it (A.B.); letizia.jannello@unimi.it (L.M.I.J.); deangeli.mario@hsr.it (M.d.A.); carolin.siech@kgu.de (C.S.); jordan.goyal@umontreal.ca (J.A.G.); zhe.tian@umontreal.ca (Z.T.); fred.saad@umontreal.ca (F.S.); pierre.karakiewicz@umontreal.ca (P.I.K.); 2Department of Neurosciences, Science of Reproduction and Odontostomatology, University of Naples Federico II, 80131 Naples, Italy; gianluigi.califano@unina.it (G.C.); roberto.larocca@unina.it (R.l.R.); marco.capece@unina.it (M.C.); nicola.longo@unina.it (N.L.); 3Martini-Klinik Prostate Cancer Center, University Hospital Hamburg-Eppendorf, 20246 Hamburg, Germany; d.tilki@uke.de; 4Department of Urology, Medical University of Graz, 8036 Graz, Austria; sascha.ahyai@medunigraz.at; 5Department of Urology, IRCCS Policlinico San Donato, 20097 Milan, Italy; luca.carmignani@unimi.it; 6Department of Urology, IEO European Institute of Oncology, IRCCS, Via Ripamonti 435, 20141 Milan, Italy; ottavio.decobelli@ieo.it (O.d.C.); gennaro.musi@ieo.it (G.M.); 7Università degli Studi di Milano, 20126 Milan, Italy; 8Division of Experimental Oncology, Unit of Urology, URI, Urological Research Institute, IRCCS San Raffaele Scientific Institute, 20132 Milan, Italy; briganti.alberto@hsr.it; 9Department of Urology, University Hospital Frankfurt, Goethe University Frankfurt am Main, 39120 Frankfurt am Main, Germany; felix.chun@kgu.de; 10Department of Urology, Comprehensive Cancer Center, Medical University of Vienna, 1090 Vienna, Austria; shahrokh.shariat@meduniwien.ac.at; 11Department of Urology, Weill Cornell Medical College, New York, NY 10065, USA; 12Department of Urology, University of Texas Southwestern Medical Center, Dallas, TX 75390, USA; 13Hourani Center of Applied Scientific Research, Al-Ahliyya Amman University, Amman 19328, Jordan; 14Department of Urology, IRCCS Ospedale Galeazzi-Sant’Ambrogio, 20157 Milan, Italy; 15Department of Oncology and Haemato-Oncology, Università degli Studi di Milano, 20122 Milan, Italy; 16Department of Urology, University Hospital Hamburg-Eppendorf, 20246 Hamburg, Germany; 17Department of Urology, Koc University Hospital, 34010 Istanbul, Turkey

**Keywords:** primary urinary tract malignant melanoma, SEER program, rare malignancy

## Abstract

**Simple Summary:**

Primary urinary tract malignant melanoma represents a rare malignancy. To date, analyses exclusively addressing contemporary diagnosed patients are unavailable. However, historical series reported lower survival rates of primary urinary tract malignant melanoma patients relative to their cutaneous counterparts. We aimed to describe the demographics, clinical characteristics, and survival outcomes of contemporary diagnosed primary urinary tract malignant melanoma patients, identified within a large-scale North American cohort.

**Abstract:**

All primary urinary tract malignant melanoma (ureter vs. bladder vs. urethra) patients were identified from within the Surveillance, Epidemiology, and End Results (SEER) database 2000–2020. Kaplan-Maier plots depicted the overall survival (OS) rates. Univariable and multivariable Cox regression (MCR) models were fitted to test the differences in overall mortality (OM). In the overall cohort (n = 74), the median OS was 22 months. No statistically significant or clinically meaningful differences were recorded according to sex (female vs. male; *p* = 0.9) and treatment of the primary (endoscopic vs. surgical; *p* = 0.6). Conversely, clinically meaningful but not statistically significant (*p* ≥ 0.05) differences were recorded according to the patient’s age at diagnosis (≤80 vs. ≥80 years old; *p* = 0.2), marital status (married 26 vs. unmarried 16 months; *p* = 0.2), and SEER stage (localized 31 vs. regional 14 months; *p* = 0.4), and the type of systemic therapy (exposed 31 vs. not exposed 20 months; *p* = 0.06). Finally, in univariable and MCR analyses, after adjustment for the SEER stage and type of systemic therapy, tumor origin within the bladder was associated with a three-fold higher OM (Hazard ratio: 3.00; *p* = 0.004), compared to tumor origin within the urethra. In conclusion, primary urinary tract malignant melanoma patients have poor survival. Specifically, tumor origin within the bladder independently predicted a higher OM, even after adjustment for the SEER stage and systemic therapy status.

## 1. Introduction

Mucosal melanoma represents a rare entity that accounts for less than 1% of all malignant melanoma cases [1]. The most frequent primary site of mucosal melanoma is in the head and neck region (55%), followed by the anus and rectum (24%), the female reproductive tract (18%), and the urinary tract mucosa (3%) [2]. Despite its rarity, mucosal melanomas behave more aggressively and have a poorer prognosis than their cutaneous counterparts [3]. 

As stated, primary urinary tract malignant melanoma represents an exceedingly rare entity [1,4,5,6]. Specifically, the urethra represents the most common tumor location, followed by the bladder, and, lastly, the ureter [7]. The first case of female urethral pigmented melanosarcoma was reported by Reed el al. in 1896 [8], while Wheelock et al. and Su et al. reported the first cases of “pigmented tumor of the bladder” in 1942 and 1962, respectively [9,10]. Conversely, the first case of ureter malignant melanoma was recorded by Judd et al. in 1962 [11].

The histogenesis of primary urinary tract malignant melanoma is unclear. To date, there are two theories: one of them supports the idea that melanoblasts migrate during embryogenesis from the neural crest into the mesenchyme, but they can also migrate in ectopic places, including the developing urinary tract, where they remain inactive for a long time and, under the influence of some local factors, they may transform into malignant cells. Another hypothesis supports the idea that urothelial cells derived from the stem cells of the urothelium may differentiate in the direction of neoplastic melanocytes [12,13,14].

There is no consensus among clinical work-up patterns to identify and stage primary malignant melanomas of the urinary tract. Regarding primary malignant melanomas located within the urethra, urethrocistoscopy, as well as magnetic resonance imaging (MRI), can be used to plan surgical treatments and evaluate the extent of soft tissue invasion in tumors located within the urethra [15]. Additionally, a computed tomography (CT) scan of the chest, abdomen, and pelvis or a whole-body positron emission tomography (PET)/CT scan with fluorodeoxyglucose (FDG) radiotracer is recommended to evaluate the spread of the disease [16,17]. Moreover, for urethral primary malignant melanomas, the American Joint Committee on Cancer Staging Manual for female urethral cancer and the staging manual for cutaneous melanomas are used [16,18]. Regarding primary malignant melanomas located within the bladder, cystoscopy with transurethral biopsy should be performed at diagnosis [6,19]. Moreover, intravenous pyelography, as well as pelvic MRI, are recommended to evaluate local invasion [19,20]. Additionally, a CT scan of the chest, abdomen, and pelvis or a whole-body FDG-PET/CT should be performed to evaluate for the spread of the disease [19,20]. Moreover, for urinary bladder malignant melanomas, a staging system has never been applied in the literature [7]. However, a distinction is made between tumors confined to the epithelium and those extending beyond it [19,20]. Regarding primary malignant melanomas located within the ureter, due to the limited number of cases, there are no guidelines on the diagnostic pattern or on the management and staging [7]. 

To date, only smaller scale and more historical population-based series have addressed urinary tract malignant melanoma cases. Indeed, the largest series available was reported by Sanchez et al., who relied on 67 urothelial melanoma patients within the Surveillance, Epidemiology, and End Results (SEER) database over a time span of 38 years (1973–2010) [21]. 

The remainder of the available studies consisted of either smaller scale or even more historical single-institutional or multi-institutional series that addressed primary urinary tract malignant melanoma cases in their entirety, regardless of tumor location [22], or focused on single tumor location [11,14,15,17,19,23,24,25,26,27,28,29]. For example, Acikalin et al. focused on primary malignant melanoma of the entire urinary tract [22]. Conversely, Oliva et al. only focused on primary malignant melanoma of the urethra [23]. Similarly, for the exceedingly rare nature of primary malignant melanoma within the bladder, only reports with, at best, two cases exist [5,14,24,25,26,27,28].

In consequence, such historical and limited data do not allow us to examine and learn from contemporary malignant melanoma cases and validates the need for a contemporary analysis, as was done in the current study. Available survival analyses comparing urinary tract melanoma patients relative to their primary cutaneous counterparts showed significantly lower five-year overall survival: 25% in the urinary tract vs. 81% in the cutaneous primary [6].

In light of these considerations, the current study addressed urinary malignant melanoma patients with a specific focus on overall mortality (OM) rates, according to established patient, treatment, and tumor characteristics. We hypothesized that these established variables may stratify OM rates. To address this hypothesis, we relied on a contemporary, large-scale North American database, such as the SEER database 2000–2020. 

## 2. Materials and Methods

### 2.1. Study Population

The SEER database samples 34.6% of the United States population in terms of demographic composition and cancer incidence [30]. Within the SEER database from 2000 to 2020, we identified patients ≥18 years old with newly diagnosed and histologically confirmed primary urinary tract malignant melanoma (International Classification of Disease for Oncology [ICD-O-3] site code C65.9, 66.9, 67.0–67.9, 68.0; ICD-O histology code 8720) [31]. We only included patients with known follow-up or vital status. All autopsy or death certificate-only cases were excluded. Since the SEER database is entirely anonymous, study-specific ethics approval was waived by the institutional review board.

### 2.2. Variables and Outcome of Interest

For each patient, the following variables of interest were recorded: age at diagnosis (years), sex, race/ethnicity (Caucasian vs. non-Caucasian), marital status (married vs. unmarried vs. unknown), site of origin (urethra vs. bladder vs. ureter), SEER stage (localized vs. regional vs. metastatic vs. unknown), treatment type of the primary (endoscopic vs. surgical), and systemic therapy status (exposed vs. non-exposed). The primary endpoint consisted of OM, according to the SEER cause of death code.

### 2.3. Statistical Analyses

Three analytical steps were completed. First, we tabulated baseline patients and tumor characteristics. Second, Kaplan-Meier plots, as well as univariable Cox regression models, addressed the OM of primary urinary tract malignant melanoma patients in different fashions according to age at diagnosis (≤80 vs. >80 years old), sex (female vs. male), marital status (married vs. unmarried), systemic therapy status (exposed vs. non-exposed), treatment type of the primary (endoscopic vs. surgical), SEER stage (localized vs. regional vs. metastatic), and site of origin (urethra vs. bladder). Third and last, multivariable Cox regression (MCR) models were fitted to address overall OM, after adjusting for site of origin (urethra vs. bladder) and known SEER stage (localized vs. regional vs. metastatic) as well as systemic therapy status (exposed vs. non-exposed). All tests were two-sided, with a significance level set at *p* < 0.05. In all statistical analyses, an R software environment for statistical computing and graphics (R version 4.1.3, R Foundation for Statical Computing, Vienna, Austria; http://www.r-project.org/, accessed on 14 January 2023) was used [32]. 

## 3. Results

### 3.1. Patient and Tumor Characteristics in Overall Primary Urinary Tract Malignant Melanoma Patients

Of all 74 primary urinary tract malignant melanoma patients, 57 (77%) were female vs. 17 (23%) male (Table 1). The median age at diagnosis was 75 years (Interquartile Ranges [IQR]: 66–82). The majority of patients were Caucasian (76%) and unmarried (49%). Additionally, 57 (77%) patients had tumors located within the urethra vs. 16 (22%) who had tumors within the bladder vs. only one (1%) within the ureter. Moreover, 35 (47%) harbored localized disease vs. 19 (26%) with regional and nine (12%) who were metastatic (Figure 1). In 11 (15%) patients, the stage was not available. Finally, of all the 66 patients who underwent treatment of the primary, 36 (55%) underwent endoscopic treatment vs. 30 (45%) who underwent surgical treatment (Figure 2). Systemic therapy was administered to 16 (22%) patients. 

### 3.2. Survival Analyses in Primary Urinary Tract Malignant Melanoma Patients

The median overall survival (OS) in the overall population of primary urinary tract malignant melanoma patients was 22 months (Figure 3). Regarding age at diagnosis, the median OS was 26 months in patients aged ≤80 years old vs. 16 months in patients aged >80 years old (∆ = 10 months) (Figure 4). Regarding sex, the median OS was 24 months in females vs. 22 months in males (∆ = 2 months). Regarding marital status, the median OS was 26 months in married patients vs. 16 months in unmarried patients (∆ = 10 months). Regarding systemic therapy exposure, the median OS was 31 months in exposed patients vs. 20 months in non-exposed patients (∆ = 11 months). Regarding treatment of the primary, the median OS was 26 months in endoscopically treated patients vs. 23 months in surgically treated patients (∆ = 3 months). Regarding SEER stage localized vs. regional vs. metastatic, the median OS was 31 months in patients with localized disease vs. 14 months in patients with regional disease (∆ = 17 months). The median OS was not reached in patients who were metastatic. In univariable Cox regression models, neither the age at diagnosis (Hazard Ratio [HR]: 1.61; 95% Confidence Interval [CI]: 0.89–2.91; *p* = 0.1), sex (HR: 1.01; 95% CI: 0.52–1.99; *p* = 0.9), marital status (HR: 1.41; 95% CI: 0.79–2.49; *p* = 0.2), systemic therapy status (HR: 2.27; 95% CI: 0.96–5.35; *p* = 0.06), or treatment of the primary (HR: 0.84; 95% CI: 0.46–1.51; *p* = 0.6) were associated with OM (Table 2).

Finally, the median OS in primary urethral malignant melanoma patients was 29 months vs. 9 months in primary bladder melanoma patients (∆ = 20 months) (Figure 5). In univariable Cox regression models, primary bladder was significantly associated with higher OM (HR: 2.90, 95% CI: 1.55–5.42; *p* < 0.001), compared to primary urethral. In the MCR model, after adjusting for known SEER stage (localized vs. regional vs. metastatic), as well as for systemic therapy status (exposed vs. non-exposed), primary bladder independently predicted a three-fold higher OM (HR: 3.00, 95% CI: 1.42–6.34; *p* = 0.004) when compared to primary urethral (Table 3). 

## 4. Discussion

Primary urinary tract malignant melanoma represents a rare entity [1,2,3,4]. No contemporary data addressing this rare entity are available. The current study aimed to fill this knowledge gap. We hypothesized that established patient, treatment, and tumor characteristics may stratify OM rates. To address this hypothesis, we relied on the SEER database 2000–2020 and made several noteworthy observations.

First, primary urinary tract malignant melanoma represents 2.8% of all mucosal melanoma cases which, in turn, represent only 1.2% of all reported melanoma cases [4,33]. In the current series, a total of 74 primary urinary tract malignant melanoma patients were identified over a period of 20 years (2000–2020). To date, the largest series addressing urinary melanoma, reported by Sanchez et al. (SEER, 1973–2010), relied on 67 urinary melanoma patients over a period of 38 years [21]. The second largest series, reported by Bishop et al. (SEER, 1988–2010), identified 2755 mucosal melanoma cases over a time span of 23 years. Of those cases, only 48 (2%) originated within the urinary tract [34]. Finally, the third largest series reported by Chang et al. (National Cancer Database [NCDB], 1985–1994), identified 1074 mucosal melanoma patients over a time span of 10 years. Of those cases, only 30 (3%) originated within the urinary tract [2]. All of the above series predominantly, if not exclusively, relied on urinary melanoma cases diagnosed prior to the year 2000. In consequence, contemporary data are clearly needed. Moreover, other previous case series addressing primary urinary tract malignant melanomas were single-institution or multi-institutional reports that relied on very limited sample sizes [5,14,15,17,19,20,22,23,24,25,26,27,28]. For example, the largest case series addressing primary urinary tract malignant melanoma regardless of tumor location was reported by Acikalin et al., who relied on eight cases [22]. Conversely, the largest series, reported by Oliva et al., focused on 15 female urethral melanoma patients [23]. Finally, exceedingly rare entities such as primary urinary bladder and primary ureter malignant melanomas were reported in case series with a maximum of two cases [26,27,29]. In light of these considerations, the rare entity of primary urinary tract malignant melanoma should ideally be investigated within the framework of large-scale population datasets such as SEER or NCDB.

Second, it is unknown whether the SEER stage predicts OM in primary urinary tract malignant melanoma. Within the current series, we recorded noteworthy differences in median OS between localized vs. regional diseases (31 vs. 14 months; ∆ = 17 months). Conversely, no calculation was possible in patients who harbored metastatic disease, as these patients did not reach median survival. However, this phenomenon can be explained by the very limited number of patients who harbored metastatic disease (n = 9). In consequence, it is impossible to ascertain the ability of current staging systems in primary urinary tract malignant melanoma. Additionally, elements required in the staging of primary cutaneous melanoma, such as Breslow thickness and the presence or absence of ulcerations, are not available for the urinary tract counterparts [35]. In consequence, this unavailability of this data may also influence staging performance in patients with urinary melanoma. To the best of our knowledge, no previous studies examined the ability of melanoma SEER staging in patients with urinary primaries. In consequence, the current data cannot be directly compared with previous studies. 

Third, we tested for differences in OM rates according to patient age at diagnosis (≤80 years vs. >80 years), sex (female vs. male), marital status (married vs. unmarried), systemic therapy status (exposed vs. non-exposed), and treatment type of the primary (endoscopic vs. surgical). In those analyses, no statistically significant or clinically meaningful differences emerged for sex (∆ = 2 months; HR: 1.01; *p* = 0.9) or treatment of the primary (∆ = 3 months; HR: 0.84; *p* = 0.6). In consequence, endoscopic vs. surgical treatment does not appear to result in clinically meaningful differences. This observation is of clinical interest. It may suggest that less debilitating endoscopic treatment may result in similar survival rates to that recorded after more extensive and debilitating treatments. However, this generalization is based on very few, distant, and far-in-between observations and should be interpreted with caution. 

Fourth, a clinically meaningful, even if not statistically significant difference was recorded in median OS, according to systemic therapy exposure, age at diagnosis, and marital status. Specifically, primary urinary tract malignant melanoma patients treated with systemic therapy exhibited a median OS of 31 months vs. 20 months in their non-exposed counterparts (∆ = 11 months). These observations indirectly validate the efficacy of systemic therapy. Similarly, patients ≤80 years old at diagnosis exhibited a median OS of 26 months vs. 16 months in patients >80 years old at diagnosis (∆ = 10 months). Interestingly, we also recorded the same difference in median OS according to marital status. Specifically, married patients exhibited a median OS of 26 months vs. 16 months in their unmarried counterparts (∆ = 10 months). In consequence, as in other primaries, more advanced age, as well as unmarried status should be considered as a potential risk factor for less favorable survival [36,37,38,39]. Specific social and clinical measures addressing or focusing on older patients and unmarried patients may potentially improve their survival [40,41,42,43]. However, due to the limited number of observations, these results cannot prompt final recommendation. Therefore, the current data should be ideally validated in other large-scale databases, such as SEER or NCDB.

Finally, when survival analyses were completed, accounting for the site of origin (urethra vs. bladder), a statistically significant and clinically meaningful difference in median OS was recorded between urethral and bladder cases (∆ = 20 months). Specifically, primary urinary tract malignant melanoma patients with primary sites within the urethra exhibited a median OS of 29 months vs. 9 months in patients with primary sites within the bladder. In the univariable Cox regression model, a site of origin within the bladder was associated with higher OM (HR: 2.90; *p* < 0.001), compared to a site of origin within the urethra. additionally, in the MCR model, even after an adjustment for the SEER stage (localized vs. regional vs. metastatic) and systemic therapy (exposed vs. not exposed), a primary site of origin within the bladder was independently associated with a three-fold higher OM (HR: 3.00; *p* = 0.004). However, the above observations regarding the differential survival benefit of primary sites should ideally be validated within an independent external validation cohort from within a different large-scale data repository, such as the NCDB.

Taken together, primary urinary tract malignant melanoma represents a very rare entity. To date, SEER staging only predicts survival rates for localized and regional primary urinary tract malignant melanoma patients. Moreover, no statistically significant or clinically meaningful differences exist according to the sex of the patient and treatment of the primary. Therefore, endoscopic treatment may be preferred to preserve body image, if survival is truly not affected. Conversely, a clinically meaningful, even though statistically insignificant difference was recorded in median OS, according to the age at diagnosis (≤80 years old: 26 months vs. >80 years old: 16 months), marital status (married: 26 months vs. unmarried: 16 months), and systemic therapy status (exposed: 31 months vs. non-exposed: 20 months). Therefore, unmarried patients, as well as patients >80 years old should be ideally counseled with greater care at initial medical decision-making and should also be followed more attentively after initial diagnosis and treatment. Moreover, systemic therapy should be recommended, as the current study has demonstrated a significant survival benefit of 11 months. This stands in contrast to a radical surgical treatment, where the benefit compared to endoscopic treatment was only 3 months. Finally, the site of the primary within the bladder independently predicted a three-fold higher OM than the urethra, even after adjustment for the SEER stage and systemic therapy status. 

The above results reflect a relatively small sample size cohort, albeit the largest one to include only patients diagnosed after 2000, and they are far from being able to draw strong recommendations. Therefore, further studies are needed to address the rare topic represented by primary urinary tract malignant melanoma.

Despite its novelty, our study is not devoid of limitations. First, SEER is a retrospective database with the potential for selection biases. However, observational databases such as SEER or NCDB represent the only opportunity to study rare primaries with robust statistical conclusions. Second, rates of local recurrence, metastatic progression, preoperative and postoperative treatments, as well as predictors of cancer-control outcomes (such as Breslow thickness and tumor ulceration) are not available in the SEER database. This limitation Is shared with other large-scale databases, such as the NCDB. Moreover, large-scale databases, such as SEER or NCDB, lack standardized specimen handling and central pathology review, which are of importance in all malignancies, including melanoma. Last but not least, large-scale databases, such as SEER or NCDB, lack details about surgical as well as systemic treatments, which are also crucial in studying the treated natural history of all malignancies, including melanoma. In consequence, our results should be tested and validated in other large-scale malignant melanoma cohorts.

## 5. Conclusions

In primary urinary tract malignant melanoma patients, the survival rate is poor. Specifically, tumor origin within the bladder was independently associated with higher OM, regardless of the SEER stage and systemic therapy status. Moreover, we also recorded clinically meaningful, even if not statistically significant benefits associated with systemic therapy administration. Similarly, marital status, as well as a younger age at diagnosis (≤80 years old), were associated with clinically meaningful higher median OS, even if not statistically significant.

## Figures and Tables

**Figure 1 cancers-15-04498-f001:**
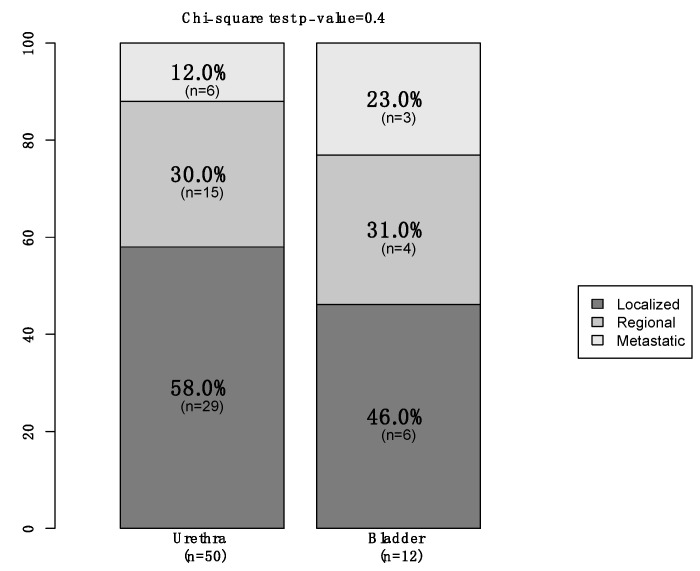
Stacked barplot depicting stage in newly diagnosed (2000–2020) primary urinary tract malignant melanoma patients, within the Surveillance, Epidemiology, and End Results (SEER) database, according to primary site (urethra vs. bladder).

**Figure 2 cancers-15-04498-f002:**
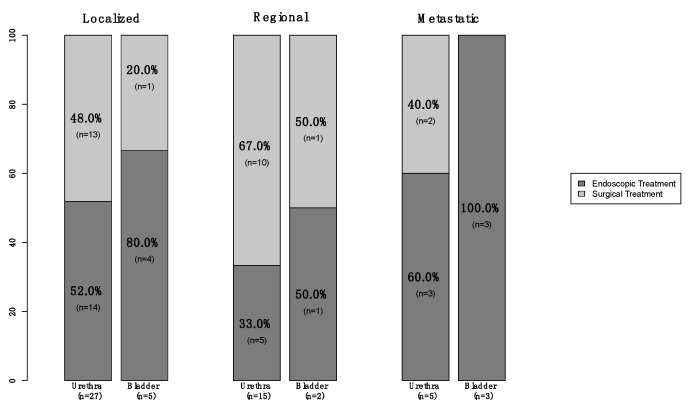
Stacked barplot depicting treatment modality (endoscopic vs. surgical) in newly diagnosed (2000–2020) primary urinary tract (urethra vs. bladder) malignant melanoma patients, within the Surveillance, Epidemiology, and End Results (SEER) database, according to known SEER stage (localized vs. regional vs. metastatic).

**Figure 3 cancers-15-04498-f003:**
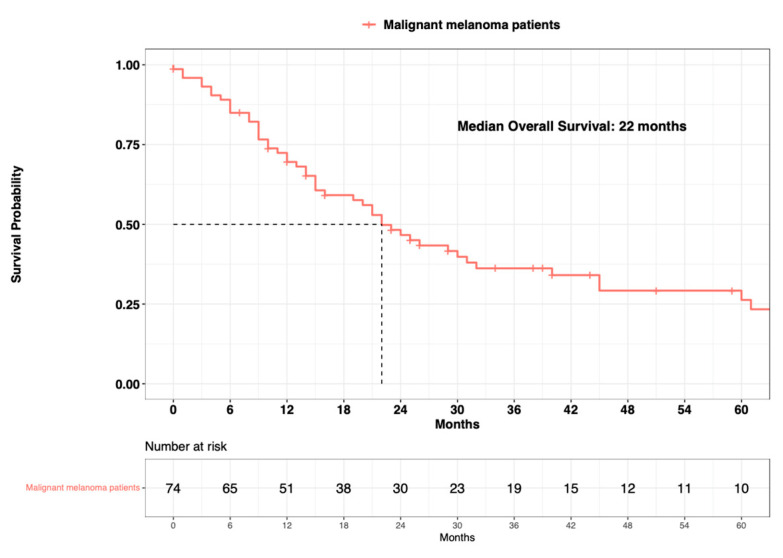
Kaplan-Meier plot depicting overall survival of 74 primary urinary tract malignant melanoma patients within the Surveillance, Epidemiology, and End Results (SEER) database 2000–2020.

**Figure 4 cancers-15-04498-f004:**
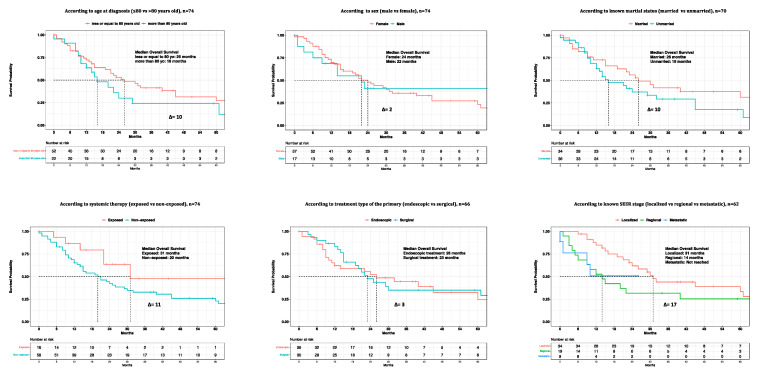
Kaplan-Meier plots depicting overall survival (OS) of primary urinary tract malignant melanoma patients within the Surveillance, Epidemiology, and End Results (SEER) database 2000–2020, according to patient (age at diagnosis, sex, known marital status), treatment (systemic therapy and treatment of the primary), and tumor (SEER stage) characteristics.

**Figure 5 cancers-15-04498-f005:**
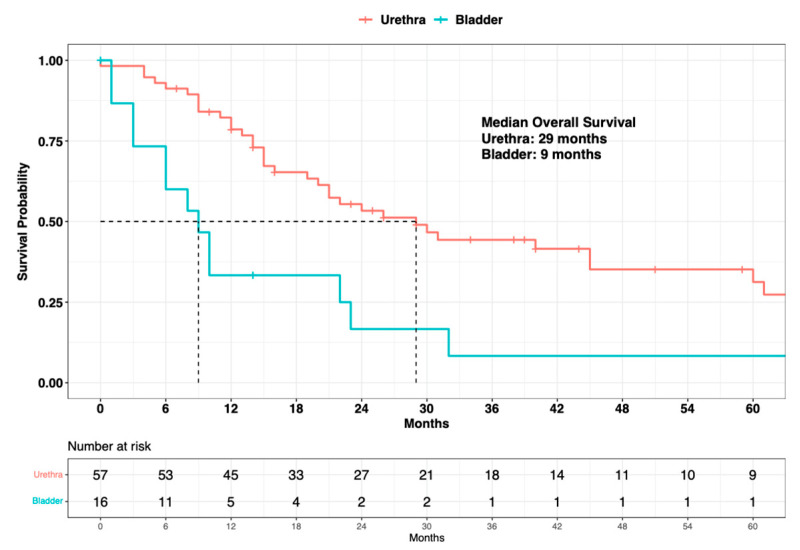
Kaplan-Meier plot depicting overall survival of 73 primary urinary tract malignant melanoma patients within the Surveillance, Epidemiology, and End Results (SEER) database 2000–2020, according to primary site (urethra vs. bladder).

**Table 1 cancers-15-04498-t001:** Baseline characteristics of newly diagnosed (2000–2020) primary urinary tract malignant melanoma patients identified within the Surveillance, Epidemiology, and End Results (SEER) database.

Characteristic	Overall, N = 74 ^1^
**Age**	75 (66, 82)
**Sex**	
*Male*	17 (23%)
*Female*	57 (77%)
**Race/ethnicity**	
Caucasian	56 (76%)
Non-Caucasian	18 (24%)
**Marital status**	
*Married*	34 (46%)
*Unmarried*	36 (49%)
*Unknown*	4 (5%)
**Tumor location**	
*Urethra*	57 (77%)
*Bladder*	16 (22%)
*Ureter*	1 (1%)
**SEER stage**	
*Localized*	35 (47%)
*Locally advanced*	19 (26%)
*Metastatic*	9 (12%)
*Unknown*	11 (15%)
**Systemic therapy**	
Yes	16 (22%)
No	58 (78%)
**Surgical treatment of the primary**	
Performed	66 (89%)
Not performed	8 (11%)

^1^ Median (IQR); n (%).

**Table 2 cancers-15-04498-t002:** Univariable Cox-regression analyses predicting Overall Mortality (OM) of newly diagnosed (2000–2020) primary urinary tract malignant melanoma (urethra vs. bladder) patients within the Surveillance, Epidemiology, and End Results (SEER) database.

Characteristic	HR ^1^	95% CI ^1^	*p*-Value
**Site of origin**			
*Urethra*	—	—	
*Bladder*	2.90	1.55–5.42	<0.001
**Age at diagnosis**			
*≤80 years*	—	—	
*>80 years*	1.61	0.89–2.91	0.1
**Sex**			
*Female*	—	—	
*Male*	1.01	0.52–1.99	0.9
**Marital status**			
*Married*	—	—	
*Unmarried*	1.41	0.79–2.49	0.2
**Systemic therapy**			
*Exposed*	—	—	
*Not exposed*	2.27	0.96–5.35	0.06
**SEER stage**			
*Localized*	—	—	
*Regional*	1.35	0.69–2.60	0.4
*Metastatic*	1.67	0.57–4.90	0.4
**Treatment of the primary**			
*Endoscopic*	—	—	
*Surgical*	0.82	0.45–1.49	0.5

^1^ HR = Hazard Ratio; CI = Confidence Interval.

**Table 3 cancers-15-04498-t003:** Multivariable Cox-regression analyses predicting overall mortality (OM) of newly diagnosed (2000–2020) primary urinary tract malignant melanoma (urethra vs. bladder) patients, within the Surveillance, Epidemiology, and End Results (SEER) database.

	Multivariable		
	HR ^1^	95% CI	*p*-Value
**Site of origin**			
Urethra	-	-	Ref
Bladder	3.00	1.42–6.34	0.004
**SEER stage**			
Localized	-	-	Ref
Regional	1.84	0.92–3.68	0.08
Metastatic	2.77	0.85–9.00	0.09
**Systemic therapy**			
Exposed	-	-	Ref
Non-exposed	2.67	0.96–7.45	0.06

^1^ HR = Hazard Ratio; CI = Confidence Interval.

## Data Availability

The data presented in this study are available in this article.

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
