# Peer review of "Demographics, Clinical Characteristics and Survival Outcomes of Primary Urinary Tract Malignant Melanoma Patients: A Population-Based Analysis"

_cancers, 2023, doi:10.3390/cancers15184498_

Round 1

Reviewer 1 Report

In this study the authors utilized SEER database to study GU melanoma. This is a rare disease, and this study represents the largest contemporary series about this disease. 

I would like to congratulate the authors for their work. The manuscript is well written, however, the conclusion should embark on the importance of systematic therapy over extensive surgeries in these cases.

Author Response

Reviewer #1:

Reviewer comment:  In this study the authors utilized SEER database to study GU melanoma. This is a rare disease, and this study represents the largest contemporary series about this disease. 

I would like to congratulate the authors for their work. The manuscript is well written, however, the conclusion should embark on the importance of systematic therapy over extensive surgeries in these cases.

Answer to reviewer: We thank the Reviewer for the comments and suggestions. We agree that the role of systemic therapy over extensive surgeries in mucosal melanomas is predominant. For this reason, we updated the discussion section, which now reads as follows:

“Conversely, a clinically meaningful, even though statistically insignificant difference was recorded in median OS, according to age at diagnosis (≤80 years old: 26 months vs. >80 years old: 16 months), marital status (married: 26 months vs. unmarried: 16 months) and systemic therapy (exposed: 31 months vs. non-exposed: 20 months). Therefore, unmarried patients, as well as patients > 80 years old should be ideally counseled with greater care at initial medical decision-making and should also be followed more attentively after initial diagnosis and treatment. Moreover, systemic therapy should be recommended, as the current study has demonstrated a significant survival benefit of 11 months. This stands in contrast to a radical surgical treatment, where the benefit compared to endoscopic treatment was only 3 months.”

Reviewer 2 Report

The manuscript entitled “Demographics, clinical characteristics and survival outcomes of primary urinary tract malignant melanoma patients: a population-based analysis” describes statistical comparison of melanoma tumors origin within the bladder and the urethra. The analysis shows that tumor origin within the bladder was associated with a three-fold higher OM compared to tumor origin within the urethra.

The results are clearly presented; however, the introduction, methods and discussion need to be improved:

1.       Further details on mucinous melanoma and urinary tract melanoma should be added to the introduction. It would be more informative to describe the staging of these rare tumours.

2.       Authors should provide a supplementary table of the samples they used with all the data for each patient. Without this data, the statistical analysis cannot be reviewed.

3.       It has been reported that the highest incidence rate is in patients over 80 years of age, regardless of sex, authors should consider to use this cut-off age in the survival analysis rather than 70 years.

4.       Among the results there are some misunderstandings such as the following: In line 144 authors mentioned: “…The median OS was not reached in patients who were metastatic.” Then in line 199: “… Within the current series, we recorded noteworthy differences in median OS between localized vs regional (31 vs 14 months; ∆=17 months). Conversely, no differences were recorded between regional and metastatic.”  If data was not reached, should not be a calculation about the differences.

5.       In the discussion section, the authors mention only three large series of cases of urinary tract melanoma, which is due to limited studies; however, there are many reviews and case series that are not mentioned in this article and that may enhance the quality of the discussion. For examples:  10.1016/j.prp.2020.153095, 10.7556/jaoa.2015.117, 10.1016/j.ajur.2022.01.003.

6.       There are discrepancies between the number of cases and the percentages in the thesis, which need to be corrected (line 190, line 192).

7.       Spelling mistakes should be corrected.

Round 2

Reviewer 2 Report

I have accepted the revised manuscript.